# Rituximab as a Treatment Option after Autologous Hematopoietic Stem Cell Transplantation in a Patient with Systemic Sclerosis

**DOI:** 10.3390/jpm11070600

**Published:** 2021-06-25

**Authors:** Paul Gressenberger, Philipp Jud, Gabor Kovacs, Sonja Kreuzer, Hans-Peter Brezinsek, Katharina Guetl, Viktoria Muster, Ewald Kolesnik, Albrecht Schmidt, Balazs Odler, Gabriel Adelsmayr, Peter Neumeister, Luka Brcic, Sabine Zenz, Kurt Weber, Thomas Gary, Marianne Brodmann, Winfried B. Graninger, Florentine C. Moazedi-Fürst

**Affiliations:** 1Division of Angiology, Department of Internal Medicine, Medical University of Graz, 8036 Graz, Austria; philipp.jud@medunigraz.at (P.J.); katharina.guetl@medunigraz.at (K.G.); viktoria.muster@medunigraz.at (V.M.); Thomas.gary@medunigraz.at (T.G.); marianne.brodmann@medunigraz.at (M.B.); 2Division of Pneumology, Department of Internal Medicine, Medical University of Graz, 8036 Graz, Austria; gabor.kovacs@medunigraz.at; 3Ludwig Boltzmann Institute for Lung Vascular Research, 8010 Graz, Austria; 4Division of Rheumatology and Immunology, Department of Internal Medicine, Medical University of Graz, 8036 Graz, Austria; sonja.kreuzer@uniklinikum.kages.at (S.K.); hans-peter.brezinsek@medunigraz.at (H.-P.B.); sabine.zenz@medunigraz.at (S.Z.); kurt.weber@medunigraz.at (K.W.); winfried.graninger@medunigraz.at (W.B.G.); florentine.fuerst@medunigraz.at (F.C.M.-F.); 5Division of Cardiology, Department of Internal Medicine, Medical University of Graz, 8036 Graz, Austria; ewald.kolesnik@medunigraz.at (E.K.); Albrecht.schmidt@medunigraz.at (A.S.); 6Division of Nephrology, Department of Internal Medicine, Medical University of Graz, 8036 Graz, Austria; balazs.odler@medunigraz.at; 7Division of General Radiology, Department of Radiology, Medical University of Graz, 8036 Graz, Austria; gabriel.adelsmayr@medunigraz.at; 8Division of Hematology, Department of Internal Medicine, Medical University of Graz, 8036 Graz, Austria; peter.neumeister@medunigraz.at; 9Diagnostic and Research Institute of Pathology, Medical University of Graz, 8036 Graz, Austria; luka.brcic@medunigraz.at

**Keywords:** systemic sclerosis, rituximab, autologous hematopoietic stem cell transplantation

## Abstract

Systemic sclerosis (SSc) is an intractable autoimmune disease characterized by vasculopathy and organ fibrosis. Autologous hematopoietic stem cell transplantation (AHSCT) should be considered for the treatment of selected patients with rapid progressive SSc at high risk of organ failure. It, however, remains elusive whether immunosuppressive therapies such as rituximab (RTX) are still necessary for such patients after AHSCT, especially in those with bad outcomes. In the present report, a 43-year-old man with diffuse cutaneous SSc received AHSCT. Despite AHSCT, SSc further progressed with progressive symptomatic heart failure with newly developed concomitant mitral and tricuspid valve insufficiency, thus the patient started on RTX 8 months after AHSCT. Shortly after initiation of RTX, clinical symptoms and organ functions ameliorated subsequently. Heart valve regurgitations were reversible after initiation of RTX treatment. Currently, the patient remains in a stable condition with significant improvement of clinical symptoms and organ functions. Reporting about therapies after AHSCT in SSc is a very important issue, as randomized controlled trials are lacking, and therefore this report adds to evidence that RTX can be considered as a treatment option in patients with SSc that do not respond to AHSCT.

## 1. Introduction

Systemic sclerosis (SSc) is an intractable autoimmune disease characterized by vasculopathy and organ fibrosis [1,2,3,4]. Despite major advances in the management of the disease, rates of mortality and morbidity are very high [1,2]. Although the underlying pathomechanisms are not fully understood, B cells are strongly involved in the pathogenesis of SSc, offering potential for therapeutic targeting of these cells [2]. For the treatment of selected patients, with rapid progressive SSc at high risk of organ failure, who are refractory to immunosuppressive therapies including B cell depletion, autologous hematopoietic stem cell transplantation (AHSCT) should be considered [2,3,4]. Nevertheless, AHSCT carries the risk of potential complications including infections, cancers, and treatment-related death [4]. It, however, remains elusive whether immunosuppressive therapies such as rituximab (RTX) are still necessary for such patients after AHSCT, especially in those with bad outcomes. We report a case of a 43-year-old man with diffuse cutaneous SSc (dcSSc) who primarily did not respond to immunosuppressive therapy with RTX and mycophenolate mofetil (MMF). Therefore, AHSCT was chosen as the next therapeutic option. The patient, however, did not respond to AHSCT. After AHSCT, re-initiation of B-cell depletion monotherapy with RTX finally led to significant improvement of the patient’s symptoms and organ functions.

## 2. Case Presentation

A 43-year-old man with a history of digital arthralgias and Raynaud’s phenomenon presented in December 2016 with puffy fingers, multiple digital necroses, and severely sclerotic areas involving the skin of the trunk and face, including sclerotic shortened frenulum. Additionally, the patient stated progressive dyspnea and reduced physical performance despite having been very athletic before. Laboratory testing revealed elevated titers of antinuclear antibodies (ANA-1:5120) and anti-Scl70 antibodies (>600.0 U/mL) with unremarkable C-reactive protein (CRP) levels. Based on the American College of Rheumatology (ACR)/European League Against Rheumatism (EULAR) criteria, the patient was diagnosed with diffuse cutaneous systemic sclerosis (dcSSc) [5]. In pleural sonography, multiple B-lines and pleural thickening of more than 3.8 mm was detected. Transthoracic echocardiography revealed a mildly reduced left ventricular ejection fraction (LVEF) of 45%. Subsequent right heart catheterization and coronary computed tomography excluded pulmonary hypertension and coronary heart disease. Magnetic resonance imaging (MRI) showed suspicious myocarditis. A myocardial biopsy was performed revealing chronic lymphocytic myocarditis with conspicuous small vessel disease, which was interpreted as an expression of the dcSSc. High-resolution computed tomography (HRCT) of the lungs revealed discreet fibrosis of both lower lobes which resulted in a diagnosis of SSc-associated interstitial lung disease (SSc-ILD). At the time of diagnosis, the patient’s modified Rodnan Skin Score (mRSS) was 12 points and spirometry revealed a forced vital capacity (FVC) of 85.7% predicted, a forced expiratory volume at one second (FEV1) of 87% predicted, a FEV1/FVC ratio of 84%, as well as a diffusing capacity for carbon monoxide (DLCO) of 62% predicted (Figure 1).

Due to active organ involvement, immunosuppressive therapy with rituximab (RTX) and mycophenolate mofetil (MMF) according to the Graz protocol [6] (Rituximab 500 mg: two intravenous infusions separated by 2 weeks every 3 months) was initiated as standard treatment with cyclophosphamide was refused by the patient. After immunosuppressive therapy and subsequent rehabilitation in May 2017, the patient’s initial symptoms improved with stable spirometry parameters and an ejection fraction of 55%; however, radiographic signs of the SSc-ILD did not improve and mRSS remained constantly at 11 points (Figure 1). Shortly after completion of the therapy, puffy fingers, acral necrosis of fingers and toes, and arthralgia of the ankles re-occurred and treatment with sildenafil was initiated. At the same time, spirometry parameters and echocardiographic findings (LVEF 50%) remained stable. In a follow-up visit in February 2018, NT-proBNP levels had increased significantly (18,287 pg/mL) accompanied by an increase of patient’s mRSS (14 points) (Figure 1) and persistent acral necrosis. Due to highly active multiple organ involvement and a high risk of disease progression, AHSCT was chosen as the next therapeutic option. In November 2018, the patient received AHSCT with a total of 2.08 × 10^6^ CD34 + cells per kilogram of bodyweight positively selected stem cells. Conditioning regimen consisted of 148 mg/kg (= 8.8 g total dose) cyclophosphamide and 7.5 mg/kg of rabbit anti-lymphocyte globulin, and after continuous G-CSF stimulation from day five after AHSCT sustained neutrophil engraftment (absolute neutrophil count >0.5 G/L) was observed. Stem cell mobilization was performed with Granulocyte Colony-Stimulating Factor (10 microgram/kg bodyweight) for four days and at day 5 Plerixafor (0.24 mg/kg) was added due to insufficient CD 34 positive cell numbers in the peripheral blood. CD34+ enrichment and cryopreservation were done using the CliniMACS system (Miltenyi Biotec B.V. & Co. KG, 51429 Bergisch Gladbach, Germany) separating CD34 positive stem cells by magnetic beads from residual bone marrow cells. Eight days after AHSCT, the patient developed neutropenic fever and an increase of inflammatory parameters. HRCT of the lungs revealed ground glass opacities in both lower lobes with a consolidation in the left lower lobe (Figure 2). Lung function and mRSS (15 points) deteriorated (FVC 44.5% predicted, FEV1 35.1% predicted, FEV1/FVC ratio 72.75%, DLCO 40% predicted) (Figure 1). A consecutive bronchoscopy due to suspected pneumonia revealed no microbial growth in bronchoalveolar lavage. Despite intensified antibiotic and antifungal treatment, symptoms and radiographic findings did not improve. Subsequent percutaneous CT-guided lung biopsy revealed epithelioid cell pneumonia without a detectable pathogen (Figure 2).

High-dose corticosteroid therapy with prednisolon (1 mg/kg body weight) was initiated and immunoglobulins were added due to secondary immune thrombocytopenia. Symptoms of epithelioid cell pneumonia improved consecutively, but this therapy was followed by a hypertensive scleroderma renal crisis (SRC) with proteinuria and a significant decline in kidney function. After initiation of angiotensin converting enzyme (ACE) inhibitor therapy, proteinuria ameliorated and hypertensive blood pressure values normalized. The patient was discharged in good condition eight weeks after AHSCT.

In April 2019, six months after AHSCT, the patient was readmitted due to unusual weakness, fatigue, and newly developed leg edema with a mRSS of 11 points (Figure 1). Laboratory testing revealed elevated NT-proBNP value of 11,189 pg/mL. Echocardiography revealed an LVEF of 32% and pericardial effusion as well as newly developed mitral and tricuspid regurgitation corresponding to acute heart failure (Figure 3).

At the same time, CT changes and spirometric lung function parameters slightly improved. Subsequently, heart failure treatment was intensified with carvedilol, thiazide diuretics, and a higher dose of ACE inhibitors, resulting in concomitant improvement of cardiac parameters. Due to further progression of the disease—despite AHSCT—the patient started on monotherapy with RTX in July 2019. In August 2019, after completion of the first cycle of RTX, clinical symptoms and spirometry parameters ameliorated noticeably (FVC 72.5% predicted, FEV1 79.2% predicted, FEV1/FVC ratio 89.8%), conversely DLCO slightly deteriorated to 38.2% predicted (Figure 1). In October 2019, after completion of the second cycle of rituximab, lung function (FVC 91% predicted, FEV1 95% predicted, FEV1/FVC ratio 84%, DLCO 44% predicted) and mRSS (1 point) further improved significantly (Figure 1). There was also a significant improvement of the echocardiographic findings to a LVEF of 52%, no mitral regurgitation and mild tricuspid regurgitation (Figure 4). Additionally, NT-proBNP levels decreased to 2289 pg/mL.

Currently (May 2021), the patient remains in a stable condition with further improvement in lung capacity (FVC 100% predicted, FEV1 96% predicted, DLCO 59% predicted, FEV1/FVC ratio 79%) and a mRSS of 0 points (Figure 1), as well as a stable LVEF of 60% and NT-proBNP level of 658 pg/mL.

## 3. Discussion

SSc is an intractable autoimmune disease characterized by tissue fibrosis including internal organ involvement such as lung, renal, and cardiac involvement, with no universally accepted disease-modifying therapy [1,2,3,4,5,6,7]. Especially interstitial lung disease (ILD) is frequently associated with disease-related morbidity and mortality in SSc patients [1]. Despite major advances in the understanding of SSc, the exact underlying pathomechanisms are not completely understood, thus finding adequate treatments is quite challenging. Treatment guidelines and experts recommend cyclophosphamide as standard treatment for SSc-related ILD; however, the efficacy of cyclophosphamide is limited, and treatment is often poorly tolerated [1]. In this regard, our patient refused treatment with cyclophosphamide. Furthermore, there is increasing use of MMF, an inhibitor of lymphocyte proliferation, which is better tolerated but has comparable efficacy to cyclophosphamide [1,2]. B cells seem to play a key role in the pathogenesis of SSc, leading to the exploration of biological drugs such as RTX targeting these cells [1,2]. RTX, a monoclonal chimeric antibody against CD20 that depletes peripheral B cells, has been reported to significantly improve skin thickness and lung function in SSc patients who are refractory to immunosuppressive therapy with cyclophosphamide [1,2,6]. Despite the use of several immunosuppressants, including the administration of RTX, further disease progression cannot be prevented in some patients.

In the present case, the patient initially responded to the administered immunosuppressive therapy. However, SSc relapsed and rapidly progressed resulting in the involvement of several internal organs. According to the recommendations of the EULAR [8] and a recent study by Henes et al. [9], AHSCT should be considered for the treatment of selected patients with rapid progressive SSc at high risk of organ failure. Therefore, the decision for AHSCT was made. However, only six months after AHSCT, the patient developed symptoms and systemic involvement of progressive SSc with acute heart failure again, which was unexpected as AHSCT is associated with a remission of disease activity lasting up to five years [10]. Additionally, DLCO remained low despite improvement of fibrotic changes in a HRCT scan of the lungs. Restarting RTX, however, finally led to significant improvement of the patient’s symptoms and activity score. Interestingly, heart valve regurgitations were reversible after rituximab treatment, as documented by echocardiography (Figure 4). To the best of our knowledge, this is the first report of heart valve involvement and reversible regurgitation after immunosuppression in SSc. RTX has beneficial effects in SSc patients, while there are no reported data on heart valve involvement. Furthermore, progression of SSc after such a short interval of AHSCT has not yet been reported.

The present case raises the issue if SSc activity can be controlled and inhibited permanently with immunosuppressive therapy. Recent data suggest that RTX exerts a significant impact on skin fibrosis with a good safety profile but has no effect on lung fibrosis [11]. Therefore, in case of residual ILD, antifibrotic therapy should be considered as recently shown by Distler et al. [12]. Another important feature of the present case is the non-response to AHSCT. Van Bijnen et al. [10] suggest that male sex, older age and an LVEF under 50% at baseline are associated with lower event free survival after AHSCT which may have contributed to the AHSCT failure in our patient. However, before AHSCT the patient’s internal organ involvement remained stable with an LVEF over 50%. A continuous immunosuppression in advance maybe could have obviated the need for AHSCT. The immunologic rationale for treating autoimmune disorders with an autologous hematopoietic stem cell transplantation is that AHSCT exerts an immunoablative effect on autoreactive B and T cell compartment, allowing for subsequent regeneration of a new (nonauto)-immune repertoire from the reinfused stem cells [13,14]. Consequently, a new and naive immune system reconstituted from the stem cell graft will re-establish immune tolerance. Although anti-lymphocyte globulin is a polyclonal antibody targeting preferential T cells, it can also induce complement-independent apoptosis of naive, activated B cells and plasma cells at clinically relevant concentrations [13,14,15]. Reconstitution of B cells has been reported to occur between 2 months and 1 year after AHSCT [14,15]. In patients with autoimmune disease, B cell homeostasis was restored with the recovery of naive B cells within several months [13,14,15]. It is plausible that reconstitution of naive B cells include clonal subpopulations that regain sensitivity against RTX treatment, and thus, AHSCT may have “sensitized” the patient to CD20 depletion therapy. In this respect, re-initiation of RTX was chosen after AHSCT, as the patient had progressive symptoms despite AHSCT and initiation of RTX treatment was seen as the last bastion at this time. Although some drugs have shown beneficial effects in treating SSc, there is a highly unmet medical need in exploring novel drug therapies. Recent data [16,17] showed clinically meaningful effects of rituximab in SSc patients and Elhai et al. [11] discussed a dose-dependent effect of rituximab which already has been hypothesized in the past [6,18].

## 4. Conclusions

In conclusion, reporting about therapies after AHSCT in SSc is a very important issue, as randomized controlled trials are lacking. AHSCT should be considered for the treatment of selected patients with rapid progressive SSc at high risk of organ failure who are refractory to immunosuppressive therapies; however, caution is given when preexisting heart failure is present, especially in male patients. This is the first case report regarding the use of RTX after AHSCT to treat SSc. The excellent response to RTX after AHSCT indicates that organ damage was reversible. In light of our observations, it seems that RTX can be considered as a treatment option in SSc patients that do not respond to AHSCT.

## Figures and Tables

**Figure 1 jpm-11-00600-f001:**
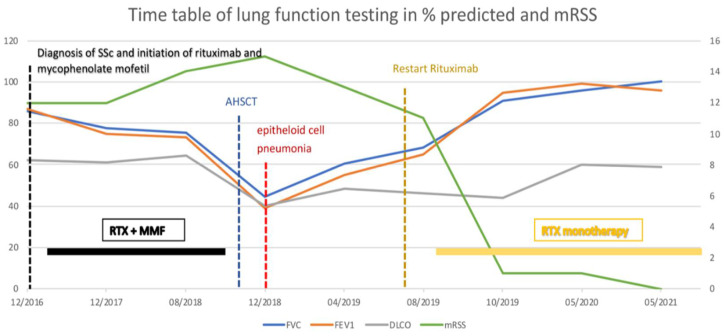
Clinical evolution of the disease showing lung function in % predicted and mRSS in relation to treatments. AHSCT = autologous hematopoietic stem cell transplantation, RTX = Rituximab, MMF = mycophenolate mofetil, mRSS = modified Rodnan Skin Score, FVC = forced vital capacity, FEV1 = forced expiratory volume during first second, DLCO = diffusing capacity for carbon monoxide.

**Figure 2 jpm-11-00600-f002:**
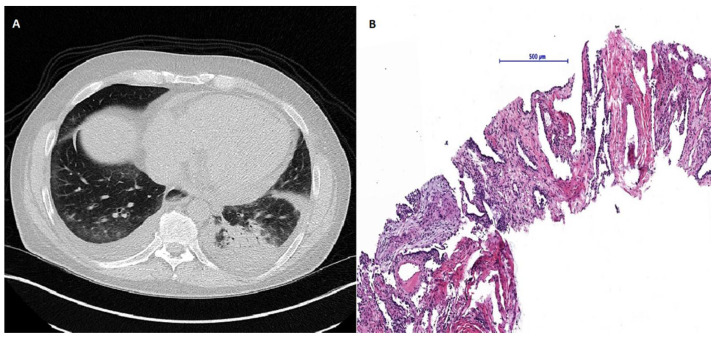
(**A**) High-resolution computed tomography of the lungs revealed ground glass opacities in both lower lobes with a consolidation in the left lower lobe. (**B**) Histological presentation of changed lung parenchyma. Needle biopsy sample of lung parenchyma showed broad interalveolar septae with fibrous tissue and chronic inflammatory cells as well as reactive changed type 2 pneumocytes. Lower left part of the sample demonstrates merging epithelioid cell granulomas without necrosis within myofibroblastic proliferation.

**Figure 3 jpm-11-00600-f003:**
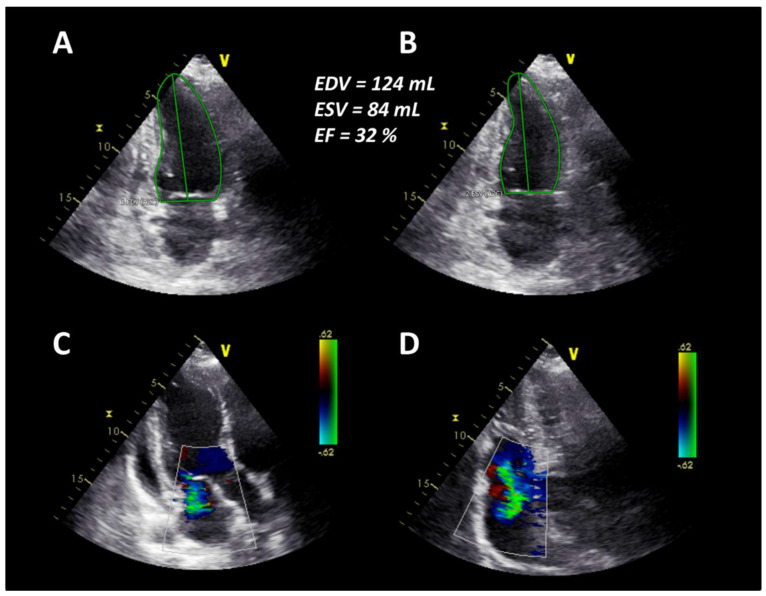
(**A**) End-diastolic borders of the left ventricle in the apical two-chamber view. (**B**) End-systolic borders of the left ventricle in the apical two-chamber view. (**C**) Mild to moderate mitral regurgitation in the apical three-chamber view. (**D**) Moderate tricuspid regurgitation in the apical four-chamber view.

**Figure 4 jpm-11-00600-f004:**
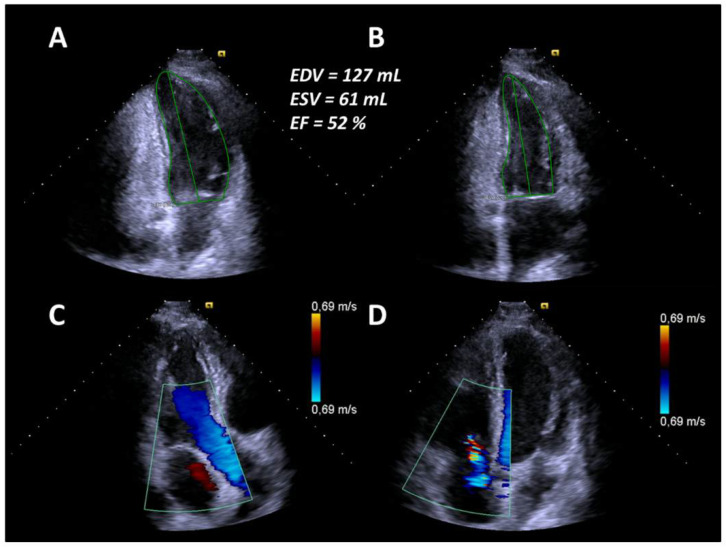
(**A**) End-diastolic borders of the left ventricle in the apical two-chamber view. (**B**) End-systolic borders of the left ventricle in the apical two-chamber view. (**C**) No mitral regurgitation in the apical three-chamber view. (**D**) Mild tricuspid regurgitation in the apical four-chamber view.

## Data Availability

The data that support the findings of this article are available from the corresponding author, [PG], upon reasonable request.

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
