# Peer review of "Rituximab as a Treatment Option after Autologous Hematopoietic Stem Cell Transplantation in a Patient with Systemic Sclerosis"

_jpm, 2021, doi:10.3390/jpm11070600_

Round 1

Reviewer 1 Report

The case report is interesting and remarkable, since it documents a successful treatment of an advanced case of Systemic Sclerosis (SSc) with rituximab, after failure from autologous stem cell transplantation.  Lacking a significant number of controlled trials in this field, this experience represents a possible useful suggestion to treat patients with SSc in advanced phase, or failing previous lines of therapy. Furthermore, the Authors proposed plausible hypothetical explanation and rationale for the successful treatment with rituximab. Interestingly, the effect of rituximab could derive from a synergistic interaction with previous stem cell transplantation.

For these reasons the case report can be accepted for publication with minor revisions.

Minor issues:

  • Lines 73-77: the Authors should better describe the details about stem cell mobilization, collection, CD34+ enrichment, and cryopreservation.
  • Line 86: the epithelioid cell pneumonia documented by lung biopsy cannot be considered primarily related to the stem cell transplantation, even if a detectable pathogen was not found.
  • The subtitles of figure 2 are confusing, maybe because there are two sets of images with the same letter (A to D).
  • Line 129: the figure 3 is not enclosed in the paper submitted.
  • Line 130: “currently, the patient remains …”, reporting the exact time from autologous stem cell transplantation, from relapse and/or from first dose of rituximab is advisable to better present the clinical evolution of the disease, in relation to the treatments.
  • The references N° 17 and 18 are not mentioned in the text.

Author Response

To the Editors of Journal of Personalized Medicine                  

Dear Editors,

Thank you for reviewing our manuscript entitled “Rituximab as a treatment option after autologous hematopoietic stem cell transplantation in a patient with systemic sclerosis”. I hereby re-submit the manuscript, which has been amended, taking into account the helpful comments of the reviewers. We have attempted to answer all questions raised and to incorporate comments. Changes to the manuscript are highlighted in red and explanations of the changes are provided below.

The contents of the manuscript, including all data and figures, have not been published or submitted for review elsewhere. All authors contributed significantly to the paper and have read and approved the currently amended version. The article is the original work of the authors.

Thank you for the honour of considering our work for publication. I am pleased to resubmit the revised manuscript to your prestigious journal. We hope that the manuscript in its revised form is now suitable for publication in Journal of Personalized Medicine.        

Sincerely yours,

Paul Gressenberger, MD

Department of Internal Medicine

Division of Angiology, Medical University of Graz

Auenbruggerplatz 15, 8036 Graz, Austria

Phone: +43 316 385 80786

Fax: +43 316 385 13788

Email: paul.gressenberger@medunigraz.at

Reviewer 1

The case report is interesting and remarkable, since it documents a successful treatment of an advanced case of Systemic Sclerosis (SSc) with rituximab, after failure from autologous stem cell transplantation.  Lacking a significant number of controlled trials in this field, this experience represents a possible useful suggestion to treat patients with SSc in advanced phase, or failing previous lines of therapy. Furthermore, the Authors proposed plausible hypothetical explanation and rationale for the successful treatment with rituximab. Interestingly, the effect of rituximab could derive from a synergistic interaction with previous stem cell transplantation. For these reasons the case report can be accepted for publication with minor revisions.

Answer: We want to thank the Reviewer for the fair and professional evaluation of our manuscript and feel pleased in light of the positive feedback. We are convinced that the issues raised by the Reviewer have helped to further improve the quality of our work.

Comments to Authors: Lines 73-77: the Authors should better describe the details about stem cell mobilization, collection, CD34+ enrichment, and cryopreservation.

Answer: We agree with the Reviewer on this point and have described stem cell mobilization, collection, CD34+ enrichment, and cryopreservation in more detail. We refer to page 3 line 85 to 95 where changes are highlighted in red.

Comments to Authors: the epithelioid cell pneumonia documented by lung biopsy cannot be considered primarily related to the stem cell transplantation, even if a detectable pathogen was not found.

Answer: We agree with the Reviewer on this point and have made the necessary corrections at page 3 line 102 to 103.

Comments to authors: The subtitles of figure 2 are confusing, maybe because there are two sets of images with the same letter (A to D).  

Answer: We agree with the Reviewer on this point and have divided figure 2 into two separate figures. We hope the changes will make the manuscript easier to understand and will not lead to further confusion. We refer to figure 3 on page 4, line 127 to 133 and figure 4  on page 5, line 148 to 155 where changes are highlighted in red. 

Comments to authors: Line 129: the figure 3 is not enclosed in the paper submitted.

Answer: We would like to apologize on this point and have made the necessary corrections. Now all figures are clearly and correct referenced in the text.

Comments to Authors: Line 130: “currently, the patient remains …”, reporting the exact time from autologous stem cell transplantation, from relapse and/or from first dose of rituximab is advisable to better present the clinical evolution of the disease, in relation to the treatments.

Answer: We agree with the Reviewer on this point and have made the necessary changes. To better present the clinical evolution of the disease, we have reported the exact time points from diagnosis of SSc to AHSCT, relapse and restart of rituximab. We refer to page 2 line 48, page 2 line 75, page 4 line 122 and page 5, line 158. We have also included a figure with spirometry parameters and modified Rodnan Skin Score in relation to the treatments over the course of time to better present the clinical evolution of the disease. The figure is referenced as figure 1 at page 2, line 62 to 69.

Comments to authors: The references N° 17 and 18 are not mentioned in the text.

Answer: We would like to thank the Reviewer for this careful observation. We have made the necessary corrections and amended the references carefully.

Reviewer 2 Report

I completed the revision of the manuscript entitled  “ Rituximab as a treatment option after autologous hematopoietic stem cell transplantation in a patient with systemic sclerosis”.

The manuscript by Gressenberger P et al. describes for the first time a case regarding the use of RTX after AHSCT treatment in patients with SSc. The authors have found that a good response to RTX following AHSCT application indicates that the organ damage is reversible. A case report is presented in a very comprehensive way and its conclusions show  a potential clinical relevance of RTX implementation into the therapeutic process. The problem of management of  SSc patients being after AHSCT treatment is a very important issue thus the results of the presented report seem to reflect the beneficial to the mentioned individuals  a good treatment option  in the case of not responding to AHSCT management.

Author Response

To the Editors of Journal of Personalized Medicine                  

Dear Editors,

Thank you for reviewing our manuscript entitled “Rituximab as a treatment option after autologous hematopoietic stem cell transplantation in a patient with systemic sclerosis”. I hereby re-submit the manuscript, which has been amended, taking into account the helpful comments of the reviewers. We have attempted to answer all questions raised and to incorporate comments. Changes to the manuscript are highlighted in red and explanations of the changes are provided below.

The contents of the manuscript, including all data and figures, have not been published or submitted for review elsewhere. All authors contributed significantly to the paper and have read and approved the currently amended version. The article is the original work of the authors.

Thank you for the honour of considering our work for publication. I am pleased to resubmit the revised manuscript to your prestigious journal. We hope that the manuscript in its revised form is now suitable for publication in Journal of Personalized Medicine.        

Sincerely yours,

Paul Gressenberger, MD

Department of Internal Medicine

Division of Angiology, Medical University of Graz

Auenbruggerplatz 15, 8036 Graz, Austria

Phone: +43 316 385 80786

Fax: +43 316 385 13788

Email: paul.gressenberger@medunigraz.at

Reviewer 2:

The manuscript by Gressenberger P et al. describes for the first time a case regarding the use of RTX after AHSCT treatment in patients with SSc. The authors have found that a good response to RTX following AHSCT application indicates that the organ damage is reversible. A case report is presented in a very comprehensive way and its conclusions show  a potential clinical relevance of RTX implementation into the therapeutic process. The problem of management of  SSc patients being after AHSCT treatment is a very important issue thus the results of the presented report seem to reflect the beneficial to the mentioned individuals  a good treatment option  in the case of not responding to AHSCT management.

Answer: We want to thank the Reviewer for the fair and professional evaluation of our manuscript and feel honored in light of the positive feedback.
